# Factors Affecting the Population of Excited Charge Transfer States in Adenine/Guanine Dinucleotides: A Joint Computational and Transient Absorption Study

**DOI:** 10.3390/biom14121548

**Published:** 2024-12-03

**Authors:** Vasilis Petropoulos, Lara Martinez-Fernandez, Lorenzo Uboldi, Margherita Maiuri, Giulio Cerullo, Evangelos Balanikas, Dimitra Markovitsi

**Affiliations:** 1Dipartimento di Fisica, Politecnico di Milano, Piazza Leonardo da Vinci 32, 20133 Milano, Italylorenzo.uboldi@polimi.it (L.U.);; 2Departamento de Química Física de Materiales, Instituto de Química Física Blas Cabrera, Consejo Superior de Investigaciones Científicas, Calle Serrano 119, 28006 Madrid, Spain; 3Istituto di Fotonica e Nanotecnologie-CNR, Piazza Leonardo da Vinci 32, 20133 Milano, Italy; 4Laboratoire d’Optique et Biosciences, Ecole Polytechnique, CNRS—INSERM, Institut Polytechnique de Paris, 91120 Palaiseau, France; 5Institut de Chimie Physique, CNRS-UMR8000, Université Paris-Saclay, 91405 Orsay, France

**Keywords:** DNA, dinucleotides, G-quadruplexes, polarity, directionality, charge transfer states, oxidative damage, photoionization, quantum yield, transient absorption spectroscopy, quantum chemistry

## Abstract

There is compelling evidence that the absorption of low-energy UV radiation directly by DNA in solution generates guanine radicals with quantum yields that are strongly dependent on the secondary structure. Key players in this unexpected phenomenon are the photo-induced charge transfer (*CT*) states, in which an electric charge has been transferred from one nucleobase to another. The present work examines the factors affecting the population of these states during electronic relaxation. It focuses on two dinucleotides with opposite orientation: 5′-dApdG-3′ (**AG**) and 5′-dGpdA-3′ (**GA**). Quantum chemistry calculations determine their ground state geometry and the associated Franck–Condon states, map their relaxation pathways leading to excited state minima, and compute their absorption spectra. It has been shown that the most stable conformer is *anti-syn* for **AG** and *anti-anti* for **GA.** The ground state geometry governs both the excited states populated upon UV photon absorption and the type of excited state minima reached during their relaxation. Their fingerprints are detected in the transient absorption spectra recorded with excitation at 266 nm and a time resolution of 30 fs. Our measurements reveal that in the large majority of dinucleotides, chromophore coupling is already operative in the ground state and that the charge transfer process occurs within ~120 fs. The competition among various relaxation pathways affects the quantum yields of the *CT* state formation in each dinucleotide, which are estimated to be 0.18 and 0.32 for **AG** and **GA**, respectively.

## 1. Introduction

Guanine radical cations, which are precursors to oxidative DNA damage, may be generated through an ionization process triggered by ion beams or electromagnetic radiation [1]. Regarding photoionization, it was long considered to be limited at wavelengths shorter than 210 nm [2,3,4]. However, during the past two decades, it has been clearly observed that DNA in aqueous solution also undergoes one-photon ionization at much longer wavelengths [5], extending to the UVB spectral domain [6]. The photoionization quantum yields (Φ_i_) at low energies are much smaller than those at high energies [7]. For example, the Φ_i_ determined for calf thymus DNA is ~50 × 10^−3^ [4,8] at 193 nm and only 2 × 10^−3^ at 266 nm [5]. Yet, the latter value is similar to the quantum yield determined at 254 nm for the totality of pyrimidine dimers [9], which are considered the major lesions provoked in DNA by direct absorption of UV radiation [10]. Therefore, the mechanism underlying the low-energy photoionization deserves further investigation.

The picture emerging from the ensemble of studies is that in low-energy photoionization, the electron is not ejected vertically, without prior geometrical rearrangement of the system, as in the case of the high-energy process [11,12,13]. Instead, photoionization is the result of a multistep mechanism, which is strongly correlated with the relaxation of the DNA electronic excited states (Figure 1). The Franck–Condon states, mostly extending over more than one nucleobase, may evolve toward excited charge transfer (*CT*) states [14,15,16,17,18,19,20,21,22,23]. In the latter, an electron is transferred between two stacked nucleobases, the driving force being the difference between their oxidation potentials, which are modulated by the local environment. A small fraction of *CT* states undergoes charge separation, which is a process that has already been reported in the literature [24,25]. Finally, an electron is ejected from the nucleobase bearing the negative charge because its ionization potential is lower compared to its neutral counterpart [26]. Thus, it is understandable why the low-energy photoionization strongly depends on the secondary DNA structure; it is not detectable for its monomeric building blocks, dinucleotides, or poorly stacked single strands (Φ_i_ < 3 × 10^−4^), and Φ_i_ increases to 1–2 × 10^−3^ for duplexes and up to 15 × 10^−3^ for guanine quadruplexes (G-quadruplexes) [5,27].

Thanks to the structural versatility of G-quadruplexes, it was shown that, among other factors, such as the nature of the metal cation in their central cavity, Φ_i_ depends on the polarity (also called directionality) of the DNA strands [27], that is the order in which nucleobases are connected via the backbone from the 5′ end to the 3′ end. Those determined for a series of structures composed of four identical 5′-XGGGG-3′ or 5′-GGGGX-3′ strands and characterized by the presence of ending groups X, [X = adenine (A) or thymine (T)] at one end, are presented on Table 1. The nearly threefold variation observed in the Φ_i_ values was correlated with the formation of *CT* states between the ending group X and guanine of the quadruplex core, G^+^→X^−^; the directionality of the transfer is determined by the fact that G is the nucleobase with the lowest oxidation potential [28]. The lifetimes of the *CT* states (τ*_CT_*), that is, the time required for charge recombination leading back to the ground state, are also presented in Table 1. They were determined for the corresponding dinucleoside monophosphates, 5′-dXpG-d3′ and or 5′-dGpdX-3′, which, for simplicity, are hereafter called dinucleotides and abbreviated as **XG** and **GX,** respectively. We remark that, indeed, the longer the lifetime, the higher the Φ_i_ value. This is in line with the mechanism depicted in Figure 1; a longer lifetime is expected to favor charge separation under the effect of conformational motions. Yet, the correlation is not linear, indicating that other factors also play an important role.

The objective of the present work is to explore the structural and electronic factors that affect the population of *CT* states in the dinucleotides **AG** and **GA**. This is a necessary step before tackling the more complex four-stranded structures, in which different types of *CT* states may be formed [32,33]. Our comparative study is conducted using quantum chemistry calculations and broadband ultrafast transient absorption spectroscopy as the main tools. Previous publications have reported lifetimes of *CT* states in these systems and estimated quantum yields (Φ*_CT_*) for their formation (Table 2). Despite the existing studies, several questions remain open. The first point concerns the stacking pattern characterizing each dinucleotide. This question was either eluded [34] or it was implicitly considered that the geometry corresponds to the conformation adopted in B-form double helices [29,35], which is determined by interactions involving the ensemble of the duplex structure. Secondly, it was considered that electronic excitations in all the stacked configurations give rise to a *CT* state; Φ*_CT_* was determined from the recovery of the ground state bleaching signal in transient absorption (*TA*) experiments [34]. In the meantime, quantum chemistry calculations revealed the existence of various Franck–Condon transitions in each dinucleotide, which may evolve toward different minima [29,31]. Moreover, quantum dynamical simulations showed that the probability of populating a *CT* state in CG steps is lower compared to that of GC steps (C = cytosine) [36]. Finally, it was estimated that the formation of the *CT* state in **AG** takes 5–10 ps; the latter conclusion was drawn from experiments whose time resolution was 150 fs after deconvolution using global analysis with multi-exponential functions [35]. However, in our recent study on the dinucleotide **TG**, which was performed with a time resolution of 30 fs, we showed that this process is much faster, occurring within 100 fs [31].

In order to elucidate the above points, in the present study, we follow the methodology we developed recently using **TG** as the showcase [31]. Transient absorption spectra (*TAS*) are recorded from 20 fs to 40 ps over the 330–650 nm range with a temporal resolution of ~30 fs. In parallel, quantum chemistry calculations determine the associated Franck–Condon states of the most stable conformers with stacked nucleobases and map their relaxation along the potential energy surfaces (PES); they also provide computed *TAS* for both the Franck–Condon states and the PES minima, facilitating their identification in the experimental data. Our work reveals the competition among various relaxation pathways, depending on the ground state geometry and the excitation wavelength, which affect Φ*_CT_*.

## 2. Materials and Methods

### 2.1. Samples

**AG** and **GA** dinucleotides, which were purified by desalting and, subsequently, using reverse phase HPLC, were purchased by Eurogentec (Liege/Belgium). The corresponding MALDI-TOF spectra, which are shown in Appendix A, indicate that both impurities and monomeric constituents have been efficiently removed. We note that electron paramagnetic resonance measurements detected other contaminants in commercially available oligonucleotides [38]. Moreover, the presence of monomeric chromophores in the solution may lead to an erroneous interpretation of the experimental results because signals originating from such “monomer impurities” are attributed to unstacked nucleobases of nucleotides. The dinucleotides were dissolved in phosphate buffer (0.12 molL^−1^, pH 7.0) using Milli-Q water. Their concentration (4 × 10^−3^ molL^−1^) was much higher than that of absorbed photons (8 × 10^−6^ molL^−1^), rendering two-photon absorption highly improbable. Experiments on the monomeric chromophores were performed using the mononucleosides 2′-deoxyadenosine (dA) and 2′-deoxyguanosine (dG) in water, instead of the corresponding mononucleotides, which have a great propensity to aggregate in solutions containing salts [39]. All measurements were performed at room temperature.

### 2.2. Spectroscopic Setups

Steady-state absorption spectra were recorded by means of a PerkinElmer Lambda 1050 Spectrophotometer, manufactured by PerkinElmer Inc., Waltham, MA, USA, utilizing quartz cells with a 1 mm path length. Circular dichroism (CD) spectra were obtained on a JASCO J-815 CD Spectrometer, produced by JASCO Corporation, Tokyo, Japan, using 0.2 mm quartz cuvettes.

*TA* experiments were conducted using an amplified Ti:Sapphire laser, Libra, Coherent, sourced from Santa Clara, CA, USA (800 nm, 100 fs pulse duration, 1 kHz repetition rate) [40]. Initially, a portion of the laser beam was frequency-doubled to drive a non-collinear optical parametric amplifier (NOPA), generating broadband visible pulses. These pulses were then compressed using chirped dielectric mirrors. Subsequently, the compressed pulses were frequency-doubled in a 20 μm-thick β-barium borate crystal to create broadband UV pump pulses that were tunable across the 250–300 nm range. The UV pump pulses were characterized by two-dimensional spectral interferometry and compressed to 24 fs (FWHM) with the aid of a prism pair before being tuned to 266 nm for the experiment. To generate broadband probe pulses, a portion of the primary laser beam was focused onto a 2 mm-thick CaF_2_ plate, producing a white light continuum that spanned from 330 nm to 650 nm. The pump and probe pulses were then non-collinearly focused onto the sample, with spot sizes of 180 μm and 95 μm, respectively. Their relative polarizations were adjusted to the magic angle (54.7°). The pump fluence was maintained at 100 μJ cm^−2^, ensuring that the differential absorption (DA) signals remained below 10^−3^. This approach effectively minimized contributions from coherent processes and solvated electrons resulting from the two-photon ionization of the solvent. To avoid photodamage during the experiment, a 6 mL solution was continuously circulated through a 1 mm-thick quartz cell using a peristaltic pump. Details on the determination of the time resolution are given in reference [31].

### 2.3. Computational Techniques

Quantum mechanical (QM) calculations were based on the density functional theory (DFT), and its time-dependent (TD-DFT) version, using the M052X functional [41,42], the 6–31G(d) basis set, and an implicit polarizable continuum model (PCM) [43] for solvent. One Na^+^ ion was considered per dinucleotide. This method is known to provide accurate results when optimizing charged species and computing spectral properties and PES in small DNA models, as well as in G-quadruplexes [20,33,44,45].

The vertical absorption energies, intensities (oscillator and rotatory strengths) of the different excited states, and corresponding PES were characterized using the above-described methodologies, but resorting to TD-DFT. The *CT* character was computed using a simple Mulliken population analysis in terms of δq, i.e., the difference between the charges in the excited state and the ground state. All these calculations were performed with the Gaussian16 program [46].

The computed spectra are compared with the experimental spectra after shifting their energy by −0.65 eV. Part of this difference arises from the absence of vibronic effects in our computations [47]. The value of −0.65 eV was chosen so that the energy computed for the lowest bright state (La) of dG in water and the corresponding value derived from deconvolution of the experimental spectrum (Figure 2 in reference [48]) coincide. A multifunctional analyzer (multiwfn program) [49] provided the transition dipole moments between the excited states for the computation of the *TAS*, for which there is no reference regarding the appropriate shift; therefore, for the sake of uniformity, we shifted them in the same way as for the steady-state spectra. In all cases, a phenomenological broadening via a Gaussian function with a width of 0.4 eV (FWHM) was applied to each transition.

## 3. Results 

### 3.1. Experimental

#### 3.1.1. Steady-State Spectroscopy

The steady-state absorption spectra of the studied dinucleotides are plotted in Figure 2a, taking into account the molar absorption coefficients (ε) provided by Eurogentec. Although they seem very similar, subtle differences can be observed; the spectrum of **AG** peak at 254 nm, and its maximum intensity is slightly higher than that of **GA**, which peaks at 256 nm. The same similarity is observed for the thermal absorption spectra (Figure 2b), which are determined from the difference between the spectra recorded at 95 °C, where stacking is destroyed, and those recorded at 23 °C. Both spectra reveal a hypochromic effect above 255 nm at room temperature, reaching 3.7 ± 0.1%, and a hyperchromic effect at shorter wavelengths.

An important difference in the interchromophore interactions within each dinucleotide is attested by the CD spectra shown in Figure 3a. The dinucleotide spectra differ from the spectrum recorded for an equivalent mixture of mononucleotides, i.e., having the same concentration of dA and dG as the dinucleotide. The CD spectrum of **GA** is characterized by well-defined positive (at 215 and 249 nm) and negative (at 204 and 270 nm) peaks. While the positive peak at 215 nm, which is preceded by a negative feature at short wavelengths, is also present in the CD spectrum of **AG**, its structure above 220 nm is ill-defined, presenting multiple local fluctuations.

#### 3.1.2. Time-Resolved Spectroscopy

The experimental *TAS* recorded from 20 fs to 100 fs for **AG**, **GA,** and an equivalent mixture of dA and dG are presented in Figure 4a–c, respectively.

The most notable feature in the evolution of the dinucleotide *TAS* is their increase in intensity over the probed spectral region. Much smaller variations are observed in the case of monomeric chromophores; the largest changes, appearing below 380 nm, are absent from the dinucleotide *TAS*. The rise in the mononucleosides’ *TA* signal at 600 nm (Figure 4d) stops at the end of the laser pulse. In contrast, in the case of the dinucleotides, it is pursued for a few tens of fs more, reaching a maximum value at around 130 fs for **AG** and 110 for **GA**. After this time, a slow decay is observed, with concomitant changes in the spectral shapes, which have been reported in our previous publication on these systems [29].

### 3.2. Computational

#### 3.2.1. Ground State Geometry

We optimized the ground state geometries considering four possible stacking modes: *anti-anti*, *anti-syn*, *syn-anti*, and *syn-syn*, where *anti* and *syn* refer to the position of each nucleobase (in the order 5′ to 3′), with respect to the angle of the glycosidic bond associated with the deoxyribose moiety. The resulting conformers are presented in Figure 5. The energies of the four conformers computed for each dinucleotide (Appendix A) are given using as a reference that of the corresponding *anti-anti* conformer, which is the one encountered in B-form duplexes.

The relative energy values reveal that the two systems do not behave in the same way. The most stable conformer of **AG** is *anti-syn*, while for **GA**, the minimum energy is found for the *anti-anti* conformer. A second important difference concerns the energy gap between the lowest and highest values: 0.18 eV for **AG** and nearly twice as high (0.33) for **GA**. Given that the energy of thermal fluctuations at room temperature amounts to 0.025 eV, the *anti-anti* conformer, whose energy differs by 0.14 eV from the next more stable one, is likely to be the dominant structure of the **GA** solution. In contrast, the *anti-syn* conformer could coexist in the **AG** solution with some percentage of the *anti-anti* conformer.

Figure 3b shows the CD spectra computed for the most stable conformers, *anti-syn* **AG** and *anti-anti* **GA**; starting from long wavelengths, both exhibit a negative peak, followed by a positive one. In contrast, the two conformers of **AG** exhibit quite opposite patterns (Figure 3c).

#### 3.2.2. Franck–Condon States and Their Evolution

In view of the abovementioned findings, we computed the properties of the excited states for *anti-syn* **AG**, *anti-anti* **AG,** and *anti-anti* **GA**. The Franck–Condon states of *anti-anti* **AG** and *anti-anti* **GA**, as well as the minima in their PES, were reported in reference [27]; their properties are also shown in the Appendix A, together with those of *anti-syn* **AG**, which were determined for the first time in the present study (Appendix A).

Based on the abovementioned data, we calculated the steady-state absorption spectrum of each conformer and simulated those of the individual transitions that compose it (Figure 6). We observe in Figure 6a that for both **AG** conformers, the maximum is located at shorter wavelengths, and its intensity is higher than that of **GA**.

The main electronic states correlated with the steady-state absorption spectrum of *anti-anti* **AG** in the 240–300 nm region are ππ*G(L_a_) (S_1_), ππ*A (S_2_), and G^+^→A^−^ *CT* (S_4_) (Figure 6b, Appendix A). Their evolution along the corresponding PES leads, respectively, to three different minima, min-ππ*G(L_a_), min-nπ*A, and min-exciton, i.e., a ππ* delocalized over A and G (Appendix A). The same Franck–Condon states, ππ*G(L_a_) (S_1_) and ππ*A (S_2_), are also present in the more stable *anti-syn* **AG** (Figure 6c, Appendix A). Although S_1_ also evolves toward a min-ππ*G(L_a_) in this conformer, S_2_ reaches a minimum with a clear G^+^→A^−^ *CT* character (δ = 0.4 a.u.) (Appendix A). In the case of *anti-anti* **AG**, a min-*CT* is reached only following the population of the S_9_ state, which is located at 218 nm according to the scale adopted in Figure 6; thus, it is very unlikely to be populated by the exciting laser pulse used in our experiments, whose spectrum is also shown in Figure 6b–d.

Four electronic transitions, whose oscillator strength ranges from 0.020 to 0.267, underlie the absorption spectrum of *anti-anti* **GA** (Appendix A and Figure 6d) [27]. Two of them, S_1_ (ππ*G(L_a_)) and S_2_ (ππ*A combined to G^+^→A^−^ *CT*; δ = 0.3 a.u.), result in a min-*CT* (δ = 0.7 a.u.) (Appendix A). S_3_, whose main character is ππ*G(L_a_), with a small contribution of G^+^→A^−^ *CT* (δ = 0.1 a.u.), results in min-ππ*G(L_a_). Finally, the S_4_ state (nπ*A) results in min-nπ*A.

Next, we calculated the TAS of the Franck–Condon states for the most stable conformers of **AG** and **GA** (Figure 7c,d), more or less overlapping with the spectrum of the laser. We observe that both the shape and the oscillator strength differ from one state to the other and from one system to the other. All these states underlie the “global” *TAS*, but their contribution depends on the probability of being populated by the excitation pulse. A rough estimate is obtained by multiplying each TAS by the oscillator strength of the corresponding steady-state transition at 266 nm (Figure 6c,d). The resulting linear combinations are shown in Figure 7c, where it appears that the intensity of the *anti-anti* **GA**
*TAS* is significantly higher than that of *anti-syn* **AG**, in particular at the high- and low-energy sides.

### 3.3. TAS of the Minima

From the computational results presented in the previous section, it appears that two main minima are expected to be populated during the excited state relaxation in the most stable conformers of **AG** and **GA**, min-*CT* and min-ππ*G(L_a_). The latter would also be reached from the less stable conformer, *anti-anti* **AG**, for which a third type of minimum, min-nπ*A (Figure 6b), may also have a non-negligible contribution, as well as, to a lesser extent, a ππ* state delocalized on both A and G (exciton).

We searched the fingerprints of min-*CT* and min-ππ*G(L_a_) in the experimental *TAS.* To this end, we performed a global analysis between 4 and 45 ps. We neglected the first 4 ps on purpose, so as to be sure that all the minima were reached and we could safely use exponential functions. As explained in detail in reference [31], if the abovementioned condition is not applied and numerous transient species coexist in the solution, the conclusions derived from such an analysis may be erroneous.

Decay-associated spectra (DAS) were derived using two exponential functions (Figure 8a,b). In order to minimize the number of variables, we fixed one of the time constants to the values determined previously from the *TA* decays measured between 500 and 645 nm by means of a different experimental setup [29]. In both cases, good-quality fits were obtained (Appendix A). The spectral profiles associated with the shortest time constant are similar for the two systems and resemble the dG *TAS* at 4 ps (Figure 8c). Despite the similarity of the two DAS, the corresponding time constants differ significantly: 2.7 ps for **GA** and 4.5 ps, which is much higher, for **AG**. This difference in the dynamics can also be directly observed in changes in the *TAS* profiles recorded for the two dinucleotides (Figure 3 and Figure 4 in reference [29]); the relative intensity of long and short wavelength peaks is inverted faster for **GA** compared to **AG**. We note that the longest time constants reported in the literature for the guanosine chromophore range from 1.9 to 2.7 ps [31,50,51,52].

The DAS associated with the longest time constants (Figure 8d) both exhibit peaks at ~384 nm and broad bands above 500 nm. A peak at 520 nm is present in the spectrum of an equimolar mixture of the guanosine radical cation and the adenosine radical anion (Figure 8d), which are determined, respectively, using ns *TA* and pulsed radiolysis [53,54]. It is not surprising that the latter does not fully overlap with the *TAS* of the *CT* states, for which the transferred charge is not complete: it is: 0.4 a.u. for *anti-syn* **AG** and 0.7 a.u. for *anti-anti* **GA**, according to our computations. The *TAS* computed for the min-*CT* of the two systems, presented in Figure 9a,b, respectively, also exhibit one peak in the UV and a second above 500 nm. Moreover, it is blue-shifted with respect to that of the corresponding Franck–Condon state.

Unlike min-ππ*G(La) and min-*CT*, it is difficult to obtain experimental information on the *TAS* of min-nπ*A. Temps and *coll*. attributed a time constant of 0.45 ps to nπ*A, which was derived from global fits with two exponential functions [55]; the same time constant has been also reported by Kwok and coll. [56]. In both studies, the presented *TAS* peak at wavelengths shorter than 310 nm. Later experiments, which were performed with lower excitation intensity and higher time resolution, reported a transient species peaking at 380 nm and characterized by a lifetime of 1.4 ± 0.2 ps instead [57]. In view of these discrepancies, the *TAS* computed for min-nπ*A in the dinucleotides are precious. That of the minor conformer *anti-anti* **AG** is shown in Figure 9a; it exhibits a rich structure over the visible domain, with an intense band in the red part located at longer wavelengths compared to those of the min-CT of *anti-syn* **AG**.

## 4. Discussion

### 4.1. Base Stacking and Photon Absorption

An early work that pioneered the study of *CT* states in dinucleotides proposed a model, according to which they should be formed only between pre-stacked nucleobases, with the remaining excitations (75% in the case of **AG**) decaying as monomer bright states [34]. Therefore, a key question in the present study is whether the nucleobases are already stacked in the ground state and to what extent. Although it is not possible to obtain a quantitative answer, our results provide important qualitative information.

The hyperchromic and hypochromic effects detected in the thermal spectra (Figure 2b) are characteristic of chromophore stacking. They result from the coupling between ππ* and *CT* states [15,58], requiring orbital overlap. Most importantly, the sharp contrast observed between the dinucleotide *TAS* and those of an equivalent mixture of mononucleotides recorded below 100 fs (Figure 4a–c) demonstrates that the percentage of non-coupled chromophores in **AG** and **GA** is low.

The relative ground state energies computed for the various conformers with stacked nucleobases (Figure 5) suggest that, while only the *anti-anti* conformer is expected to be present in **GA** room temperature solutions, the situation is more complex for **AG**. For the latter, in addition to the most stable *anti-syn* conformer, the presence of a small amount of the *anti-anti* conformer is also possible. This is in agreement with the picture emerging from the CD spectra. As previously noted, the experimental CD spectrum of **AG**, unlike that of **GA**, exhibits multiple fluctuations above 220 nm (Figure 3a). This is readily explained by the coexistence of a major (*anti-syn*) and a minor (*anti-anti*) conformer, whose computed CD spectra are characterized by opposite patterns (Figure 3c); given that the signatures of the two **AG** conformers are not canceled out in the experimental CD spectra and only fluctuations are observed, the *anti-anti* **AG** concentration in the solutions is significantly smaller compared to that of *anti-syn* **AG**.

It is also worth noting that the peaks of the steady state computed for the major and minor **AG** conformers are located at shorter wavelengths, and their intensity is higher compared to that of *anti-anti* **GA** (Figure 6a). They follow a tendency similar to that of the experimental spectra (Figure 2a) but are significantly amplified. This suggests that although the comparison between computed and experimental spectra in the present study is not intended to be quantitative, the spectra of different systems computed following the same methodology reveal interesting trends.

### 4.2. Relaxation Dynamics: From the Franck–Condon States to the Minima

A quantitative analysis of the spectra in Figure 4a,b is not possible due to three reasons. The first is the existence of several Franck–Condon states per conformer evolving toward different minima. The second is the lack of theoretical information regarding the dynamic patterns followed in this evolution. The third is the existence in **AG** of a minor conformer with an unknown concentration. However, we can make qualitative observations, assuming that the dominant structures are *anti-syn* for **AG** and *anti-anti* for **GA**, as determined theoretically, while for the former system, a smaller contribution from the *anti-anti* conformer is also possible.

The important changes observed in the *TAS* presented in Figure 4a,b are associated with the evolution along the various PES. This is supported by the rise of the *TA* signals in Figure 4d; while that of the monomers stops just after the end of the exciting laser pulse, in the case of the dinucleotides, it lasts longer. The fact that the maximum value is reached faster for **GA** compared to the **AG** (110 fs vs. 130 fs) could be due to an equilibrium between two conformers.

Although we cannot rule out that the relaxation process has already started before the end of the laser pulse, the *TAS* at 30 fs (Figure 7d) are likely to be dominated by the Franck–Condon states. The stronger intensity observed for **GA** is indeed reproduced in the *TAS* computed for the most stable conformers of the two dinucleotides (Figure 7c), taking into account the probability of each Franck–Condon to be populated by the exciting laser pulse (Figure 6).

Next, we search in the *TAS* evolution the fingerprints of the minima predicted theoretically: sy: min-ππ*G(L_a_), min-*CT*, and min-nπ*A. The global analysis performed on the experimental *TAS* recorded from 4 to 45 ps confirmed that min-ππ*G(L_a_) and min-*CT* are indeed populated in both **AG** and **GA** (Figure 8). The early time dynamics (Figure 4) show that this population is very fast.

Starting from the simpler case of **GA**, for which a single conformer is dominant, we observe that, in parallel with the intensity increase, the band initially peaking around 620 nm progressively shifts to shorter wavelengths, reaching 580 nm at 100 fs (Figure 4b). We assign these changes to the evolution along the PES, leading from S_1_ and S_2_ to min-*CT*. According to the computed *TAS* (Figure 9b), the population of min-*CT* is manifested indeed by an increase in intensity and a blue shift compared to the corresponding Franck–Condon state. Consequently, we deduce that the changes observed in the long-wavelength band in Figure 4b reflect the dynamics of the charge transfer process.

An important difference between the *TAS* of the two systems is that the long-wavelength band, shifting to shorter wavelengths for **GA**, progressively splits into two components for **AG**; at 100 fs, a peak at 586 nm and a shoulder at ~530 can be distinguished. Such a complex behavior is assigned to the population of min-nπ*A in the minor *anti-anti* **AG** conformer. Our interpretation is based on the computed *TAS*; the low-energy band observed in that of min-nπ*A is red-shifted compared to that of the min-*CT* of *anti-syn* AG (Figure 9a). Thus, the concomitant population of these two minima should induce a spectral evolution similar to that observed in Figure 4a.

The peak at ~440 nm growing in the *TAS* of both **AG** and **GA** is correlated with the guanosine chromophore [31]. Several experimental studies reported that the excited state relaxation of this chromophore in solution is very complex [50,51,52,59,60]. Moreover, a theoretical work that was performed combining quantum chemistry calculations and molecular dynamics simulations identified the presence of three different minima in the PES of its first singlet excited state [61]. The first minimum absorbs at longer wavelengths (around 400 nm) than the third one, which is characterized by the longest lifetime. The latter was indeed detected through our global analysis (Figure 8c).

### 4.3. Quantum Yields

In our previous publication, we reported on the relative Φ*_CT_* values; the value for **GA** was found to be 75% higher than that for **AG** [29]. This conclusion was drawn from the relative intensities of the UV bands present in the *TAS* at zero time, which were obtained from the *TAS* at 15 ps according to the equation (DA)_0_ = (DA)_15ps_/exp(–15ps/τ*_CT_*). In the present more refined study, we examine this issue again. Now, we start from the *TAS* at 40 ps and focus on the low-energy band, which has been associated with the sum of the absorption spectra of the dA radical anion and the dG radical cation (Figure 8d). Applying the same formula as before, we find that the peak intensity is 71% higher for **GA** than for **AG** (Figure 10a). Then, using the molar absorption coefficient corresponding to an equimolar mixture of the dA radical anion [53] and the dG radical cation [54] (1230 mol^−1^Lcm^−1^) and considering the concentration of absorbed photons (8 × 10^−6^ molL^−1^), we obtain a Φ*_CT_* value of 0.18 for **AG** and 0.32 for **GA**. According to our calculations, the oscillator strength of the low-energy peaks present in the *TAS* of the min-*CT* is higher than that corresponding to the sum of the absorption spectra computed for the dA radical anion and the dG radical cation (Figure 10b), suggesting that the determined values represent upper limits for the Φ*_CT_*.

## 5. Conclusions

Our present theoretical and experimental study on the **AG** and **GA** dinucleotides, which is the continuation of our previous work on these systems [29], sheds important light on the excited state relaxation occurring in these systems upon excitation at 266 nm. Based on a series of quantum chemistry calculations, combined with broad-band transient absorption spectroscopy at the exceptional resolution of 30 fs, we followed the processes occurring at early times, from photon absorption to the population of the *CT* states.

The main findings are summarized as follows:➢Our computations on these systems showed that the stacking pattern corresponding to the most stable geometry of the studied systems depends on the dinucleotide polarity; while the *anti-anti* conformation present in B-form duplexes is adopted in **GA**, **AG** corresponds to the *anti-syn* configuration.➢The computed Franck–Condon states and their evolution depend both on the polarity and the stacking geometry.➢The excited state minima predicted theoretically, min-*CT*, min-ππ*G(La), and min-nπ*A, were detected in the experimental *TAS*; they are reached within ~120 fs.➢From the experimental *TAS*, we deduced that the largest portion of the nucleobases within the dinucleotides are electronically coupled in their ground state. This does not preclude the existence of a minimum localized on a single nucleobase, min-ππ*G(La), whose lifetime depends on the dinucleotide polarity.➢As a result of the competition among the different relaxation paths, the quantum yields determined for the formation of the *CT* state are relatively low. The estimated upper values in **AG** and **GA** are 0.18 and 0.32, respectively.➢From a methodological point of view, our computations showed that the *TAS* of the various excited states involved in the relaxation process exhibit different spectral shapes and oscillator strengths. This reason renders the use of the pre-exponential factors derived from fits of the *TA* signals inappropriate for the determination of their populations.

The knowledge acquired by the present study on dinucleotides will serve as a foundation for the characterization of the abovementioned processes in G-quadruplexes, with adenines at the 3′ and/or the 5′ ends, whose Φ_i_ values at 266 nm are the highest determined for any DNA system [27].

## Figures and Tables

**Figure 1 biomolecules-14-01548-f001:**
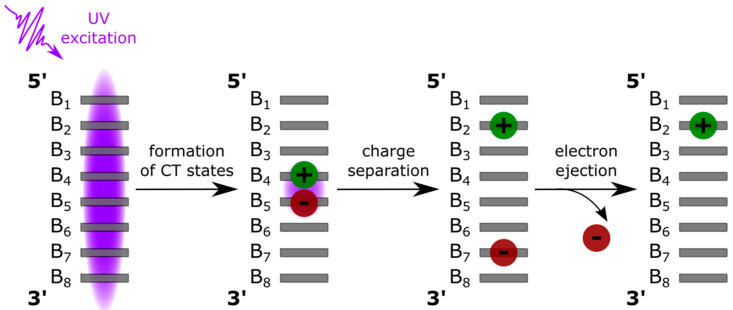
Successive steps potentially leading to DNA photoionization at low energies. B_i_ designates stacked nucleobases.

**Figure 2 biomolecules-14-01548-f002:**
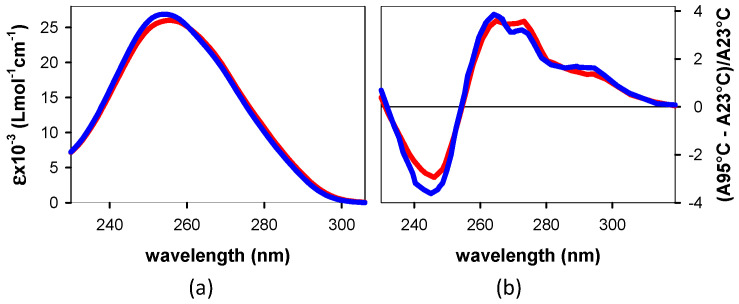
(**a**) Steady-state and (**b**) thermal absorption spectra determined for **AG** (blue) and **GA** (red). The thermal spectra are the difference between the steady-state absorption spectra recorded for each dinucleotide at 95 °C (A95 °C) and 23 °C (A23 °C), divided by A23 °C.

**Figure 3 biomolecules-14-01548-f003:**
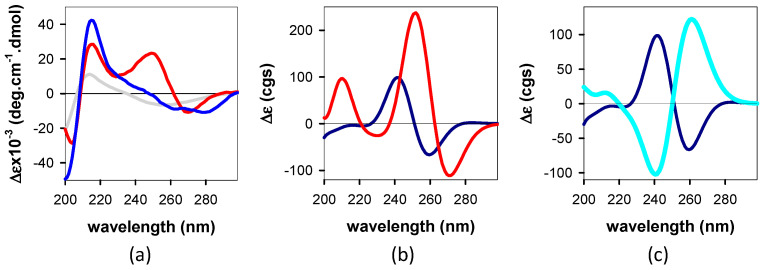
CD spectra recorded for **AG** (blue), **GA** (red), and an equimolar mixture of dA and dG (gray) (**a**) and computed for *anti-syn* **AG** (dark blue), *anti-anti* **GA** (red), and *anti-anti* **AG** (cyan) (**b**,**c**).

**Figure 4 biomolecules-14-01548-f004:**
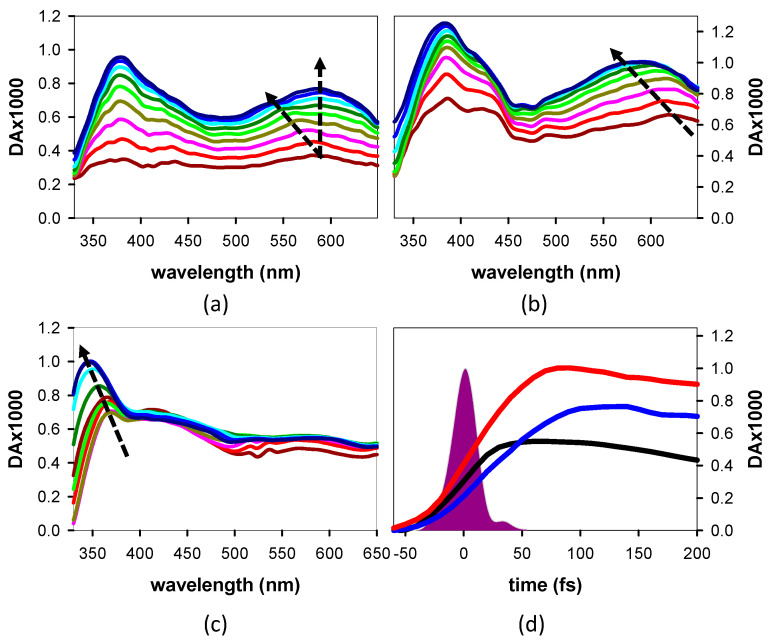
Experimental *TAS* obtained for **AG** (**a**), **GA** (**b**), and an equivalent mixture of dA and dG (**c**) recorded between 20 fs (dark red) and 100 fs (dark blue) with 10 fs steps. *TA* signals at 600 nm obtained for **AG** (blue), **GA** (red), and an equivalent mixture of dA and dG (black) (**d**).

**Figure 5 biomolecules-14-01548-f005:**
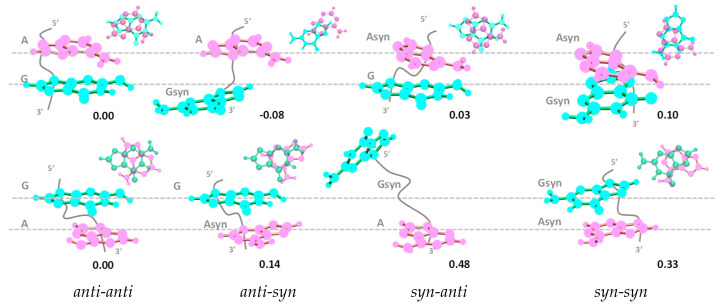
Side and top views of the ground state structures corresponding to the four possible conformers with stacked nucleobases computed for **AG** and **GA**. Their relative energies (ΔG, in eV) are shown in black. Adenine and guanine are depicted in pink and turquoise, respectively, and the backbone is in gray. The top view of *syn-anti* **GA** has been omitted because the overlap is very poor.

**Figure 6 biomolecules-14-01548-f006:**
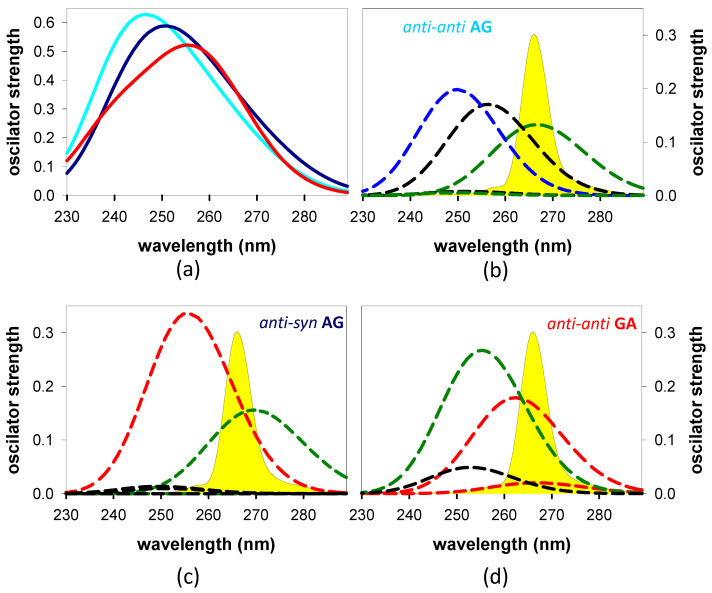
Steady-state absorption spectra (solid lines) computed for *anti-anti* **AG** (cyan), *anti-syn* **AG** (dark blue), and *anti-anti* **GA** (red) (**a**). Individual transitions (dashed lines, see Appendix A) composing each one of them (**b**–**d**); the color code of the dashed lines is defined by the minimum toward which evolves each individual transition (see Appendix A): min-*CT* (red); min ππ*G(La) (green); min-nπ*A (black); ππ* exciton delocalized on A and G (blue). The spectrum of the exciting laser pulse is shown in yellow.

**Figure 7 biomolecules-14-01548-f007:**
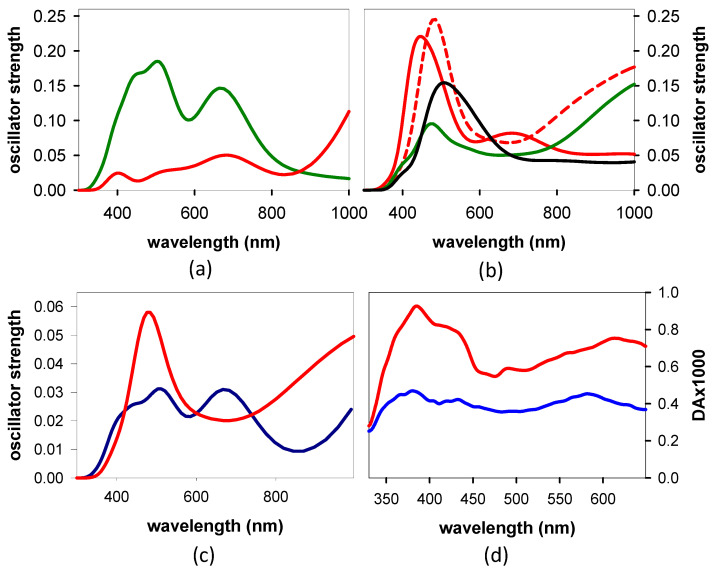
*TAS* of Franck–Condon states. Computed for S_1_ (green) and S_2_ (red) of *anti-syn* **AG** (**a**) and S_1_ (red solid line), S_2_ (red dashed line), S_3_ (green), and S_4_ (black) of *anti-anti* **GA** (**b**); the color code is the same as that in Figure 6c,d. (**c**). Weighted sum of the *TAS* of all the states in Figure 6c (*anti-syn* **AG**: dark blue) and Figure 6d (red: *anti-anti* **GA**), each one scaled by the corresponding oscillator strength at 266 nm. (**d**) Experimental *TAS* recorded at 30 fs for **AG** (blue) and **GA** (red).

**Figure 8 biomolecules-14-01548-f008:**
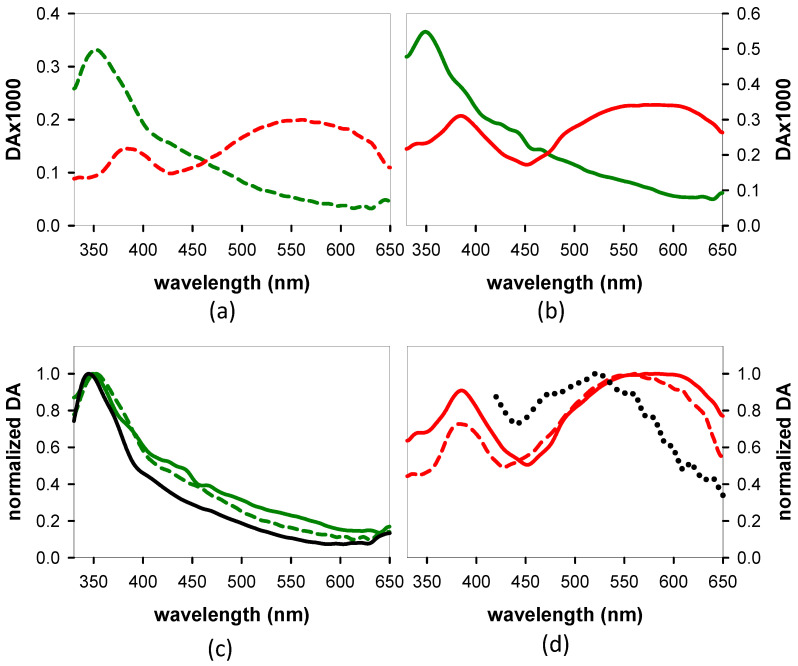
DAS derived from global fits of the *TAS* of **AG** (dashed lines, (**a**)) and **GA** (solid lines, (**b**)) between 4 and 45 ps. Red: DAS associated with the longer lifetime, 112 ps for **AG** and 170 ps for **GA**; green: DAS associated with the shorter lifetime, 4.5 ps for **AG** and 2.7 ps for **GA**; solid black line in (**c**): *TAS* obtained for dG alone at 4 ps; dotted line in (**d**): absorption spectrum corresponding to an equimolar mixture of the adenosine radical anion [53] and the guanosine radical cation [54]. The *TAS* in (**c**) are normalized at their maximum intensity and in (**d**) at 520 nm.

**Figure 9 biomolecules-14-01548-f009:**
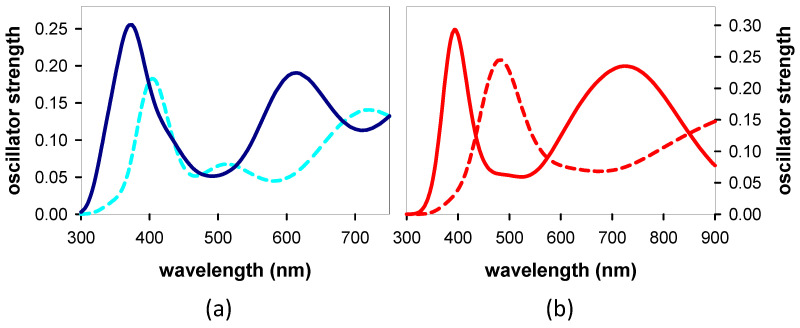
Comparison of computed *TAS*: (**a**) min-*CT* of *anti-syn* **AG** (dark blue line) and min-nπ*A of *anti-anti* **AG** (cyan dashed line); (**b**) min-*CT* of *anti-anti* **GA**: S_2_ Franck–Condon state (dashed line) evolving toward the min-*CT* (solid line).

**Figure 10 biomolecules-14-01548-f010:**
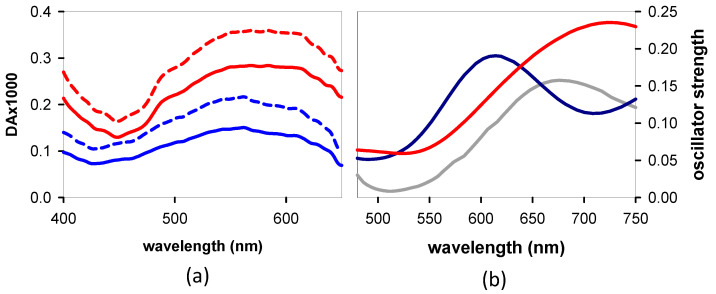
Low-energy band of the *CT* states. (**a**) *TAS* recorded for **AG** (blue) and **GA** (red) at 40 ps (solid lines) and extrapolated to zero time (dashed lines). (**b**) *TAS* computed for the *CT*-min of *anti-syn* **AG** (dark blue) and *anti-anti* **GA** (red); the gray line corresponds to the sum of the spectra computed for the dA radical anion and the dG radical cation (average for the *anti* and *syn* conformations for each radical).

**Table 1 biomolecules-14-01548-t001:** Correlation between the Φ_i_ values determined for tetramolecular G-quadruplexes composed of four identical strands 5′-XGGGG-3′/5′-GGGGX-3′ (X = adenine, thymine) and the lifetimes of the *CT* states (τ*_CT_*) in **GX** and **XG** dinucleotides; excitation wavelength: 266 nm.

G-Quadruplex ^1^	Φ_i_ × 10^3^ [27]	Dinucleotide ^2^	τ*_CT_* (ps)
4 × (5′-GGGGA-3′)	12.6 ± 0.1	**GA**	170 ± 10 [29]
4 × (5′-AGGGG-3′)	9.9 ± 0.4	**AG**	112 ± 12 [29]
4 × (5′-GGGGT-3′)	8.8 ± 0.2	**GT**	13 ± 1 [30]
4 × (5′-TGGGG-3′)	4.7 ± 0.7	**TG**	5.44 ± 0.03 [31]

^1^ From ns *TA* experiments; ^2^ from fs *TA* experiments.

**Table 2 biomolecules-14-01548-t002:** Lifetimes (in ps) of the *CT* states in **AG** and **GA** dinucleotides reported in the literature; in parentheses: excitation/probed wavelengths or energies.

AG	GA
105 ± 30 (267 nm/252 nm) ^1^ [34]	-
124 ± 4 (260 nm/330–680 nm) ^2^ [35]	-
280 ± 160 (266 nm/1500–1700 cm^−1^) ^3^ [37]	420 ± 120 (266 nm/1500–1700 cm^−1^) ^4^ [37]
112 ± 12 (266 nm/330–680 nm) [29]	170 ± 10 (266 nm/500–645 nm) [29]

Φ*_CT_*: ^1^ 33 ± 6%; ^2^ >26%; ^3^ 32 ± 15%; ^4^ 42 ± 20%.

## Data Availability

The original contributions presented in the study are included in the article/Appendix A, further inquiries can be directed to the corresponding authors.

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
