# Peer review of "Factors Affecting the Population of Excited Charge Transfer States in Adenine/Guanine Dinucleotides: A Joint Computational and Transient Absorption Study"

_biomolecules, 2024, doi:10.3390/biom14121548_

Round 1
Reviewer 1 Report
Comments and Suggestions for Authors
I reviewed the manuscript entitled: “Factors affecting the population of excited charge transfer states in adenine/guanine dinucleotides: a joint computational and transient absorption study” submitted to Biomolecules. This paper describes the experimental and theoretical study about the photoexcited states of nucleotides. The transient absorption, steady state absorption, and CD spectral measurements and DFT and TD-DFT calculations were performed to reveal the initial process of photoexciting stacked nucleotide dimers and their relaxation processes through charge-transfer state. This is an interesting study in the nucleotide photochemistry. The minor noticed points are listed below.
1) Does the first step of Figure 1 show the delocalization of excited energy or the photoexcitation of nucleotide ensemble? If so, a short explanation in the text is helpful for readers.
2) Introduction, CT state: The positive nucleotide and the negative nucleotide should be explained. The redox potential values of nucleotides may be an important data. For this purpose.
3) Figure 4: In this figure, the difference between the structure of AG and that of GA is unclear.
Author Response
- Does the first step of Figure 1 show the delocalization of excited energy or the photoexcitation of nucleotide ensemble? If so, a short explanation in the text is helpful for readers.
- Introduction, CT state: The positive nucleotide and the negative nucleotide should be explained. The redox potential values of nucleotides may be an important data. For this purpose.
In answer to questions 1 and 2, we added the following explanation in the “Introduction”:
"The Franck-Condon states, mostly extending over more than one nucleobase, may evolve toward excited charge transfer (CT) states [1-10]. In the latter, an electron is transferred between two stacked nucleobases, the driving force being the difference between their oxidation potentials, which are modulated by the local environment. A small fraction of CT states undergoes charge separation, a process already reported in the literature [11,12]. Finally, an electron is ejected from the nucleobase bearing the negative charge, because its ionization potential is lower compared to its neutral counterpart [13].
… formation of CT states between the ending group X and a guanine of the quadruplex core, G+®X-; the directionality of the transfer is determined by the fact that G is the nucleobase with the lowest oxidation potential [14]."
We avoided citing precise values for the oxidation potentials because they are modulated by the secondary structure and the local environment.
- Figure 4: In this figure, the difference between the structure of AG and that of GA is unclear.
We modified Figure 4, which is Figure 5 in the revised manuscript.
references
- Nogueira, J.J.; Plasser, F.; Gonzalez, L. Electronic delocalization, charge transfer and hypochromism in the UV absorption spectrum of polyadenine unravelled by multiscale computations and quantitative wavefunction analysis. Chem. Sci. 2017, 8, 5682-5691.
- Plasser, F.; Aquino, A.; Lischka, H.; Nachtigallová, D. Electronic Excitation Processes in Single-Strand and Double-Strand DNA: A Computational Approach. Top. Curr. Chem. 2015, 356, 1–38.
- Spata, V.A.; Matsika, S. Role of Excitonic Coupling and Charge-Transfer States in the Absorption and CD Spectra of Adenine-Based Oligonucleotides Investigated through QM/MM Simulations. J. Phys. Chem. A 2014, 118, 12021-12030.
- Santoro, F.; Improta, R.; Avila, F.; Segado, M.; Lami, A. The interplay between neutral exciton and charge transfer states in single-strand polyadenine: a quantum dynamical investigation. Photochemical & Photobiological Sciences 2013, 12, 1527-1543.
- Aquino, A.J.A.; Nachtigallova, D.; Hobza, P.; Truhlar, D.G.; Hattig, C.; Lischka, H. The Charge-Transfer States in a Stacked Nucleobase Dimer Complex: A Benchmark Study. J. Comput. Chem. 2011, 32, 1217-1227.
- Blancafort, L.; Voityuk, A.A. Exciton delocalization, charge transfer, and electronic coupling for singlet excitation energy transfer between stacked nucleobases in DNA: An MS-CASPT2 study. J. Chem. Phys. 2014, 140.
- Improta, R.; Santoro, F.; Blancafort, L. Quantum Mechanical Studies on the Photophysics and the Photochemistry of Nucleic Acids and Nucleobases. Chem. Rev. 2016, 116, 3540-3593.
- Chen, J.; Zhang, Y.; Kohler, B. Excited states in DNA strands investigated by ultrafast laser spectroscopy. Top. Curr. Chem. 2015, 356, 39-87.
- Schreier, W.J.; Gilch, P.; Zinth, W. Early Events of DNA Photodamage. Annu. Rev. Phys. Chem. 2015, 66, 497-519.
- Kwok, W.M.; Ma, C.S.; Phillips, D.L. "Bright" and "Dark" excited states of an alternating AT oligomer characterized by femtosecond broadband spectroscopy. J. Phys. Chem. B 2009, 113, 11527-11534.
- Bucher, D.B.; Pilles, B.M.; Carell, T.; Zinth, W. Charge separation and charge delocalization identified in long-living states of photoexcited DNA. Proc. Natl. Acad. Sci. U.S.A. 2014, 111, 4369-4374.
- Kufner, C.L.; Crucilla, S.; Ding, D.; Stadlbauer, P.; Sponer, J.; Szostak, J.W.; Sasselov, D.D.; Szabla, R. Photoinduced charge separation and DNA self-repair depend on sequence directionality and stacking pattern. Chem. Sci. 2023.
- Schiedt, J.; Weinkauf, R.; Neumark, D.M.; Schlag, E.W. Anion spectroscopy of uracil, thymine and the amino-oxo and amino-hydroxy tautomers of cytosine and their water clusters. Chem. Phys. 1998, 239, 511-524.
- Palecek, E.; Bartosik, M. Electrochemistry of Nucleic Acids. Chem. Rev. 2012, 112, 3427-3481.

Reviewer 2 Report
Comments and Suggestions for Authors
This combined theoretical (DFT, TD-DFT) and experimental study of excited states in dinucleotides made up one each of G and A elucidates the production of excited states including charge transfer states. The multiplicity of conformation structures for each dinucleotide are treated in computation studies and the more stable structures employed in excited state calculations. The work leads to interesting experimental and theoretical findings. The authors need to consider a few comments below.
Comments:
1. The authors state the work is on dinucleotides of G and A which could include two phosphates in the structure. No clear structure of the GA and AG systems but the chemical designation 5’-dApdG-3’ for (AG) and 5’-dGpdA-3’ (GA). These have only one phosphate and therefore are dinucleotide monophosphates.
2. Figure S1 The MALDI TOF spectra of AG and GA provided by Eurogen are blurred and the details can not be read.
3. TD-DFT with the M052X functional, the 6-31G(d) basis set and PCM solvent model are employed. These are adequate for the calculations except that it is important the PCM model use non equilibrium model for excited state calculations. For the PCM model this is done automatically for TD DFT calculations. However, as the state equilibrates as is done in this work the solvent model would have to become the equilibrium model. The time frame for the formation of the charge transfer state becomes important in this regard. A discussion of this is needed with the expected effects of the solvent reorganization, for example, in Figure 8a b does the band shift include solvent reorganization.
Author Response
Comments:
- The authors state the work is on dinucleotides of G and A which could include two phosphates in the structure. No clear structure of the GA and AG systems but the chemical designation 5’-dApdG-3’ for (AG) and 5’-dGpdA-3’ (GA). These have only one phosphate and therefore are dinucleotide monophosphates.
We thank the reviewer for this remark. “Dinucleotide” is indeed a simplified term. We made the precision in the Introduction:
…dinucleoside monophosphates 5’-dXpG-d3’ and or 5’-dGpdX-3’, which for simplicity are called hereafter just dinucleotides and are abbreviated as XG and GX, respectively.
- Figure S1 The MALDI TOF spectra of AG and GA provided by Eurogen are blurred and the details can not be read.
The quality of the Figure is limited by the quality of pdf files provided by Eurogentec. However, we managed to improve it slightly.
TD-DFT with the M052X functional, the 6-31G(d) basis set and PCM solvent model are employed. These are adequate for the calculations except that it is important the PCM model use non equilibrium model for excited state calculations. For the PCM model this is done automatically for TD DFT calculations. However, as the state equilibrates as is done in this work the solvent model would have to become the equilibrium model. The time frame for the formation of the charge transfer state becomes important in this regard. A discussion of this is needed with the expected effects of the solvent reorganization, for example, in Figure 8a b does the band shift include solvent reorganization.
We thank the reviewer for giving us the opportunity to clarify this issue. Indeed, the choice between the nonequilibrium and the non-equilibrium model is not simple in the present study. On one hand, we move from a few tenths of fs (when the nonequilibrium model is more appropriate) to tenths of ps, time on which the density should be already equilibrated. On the other hand, previous calculations were done with nonequilibrium and, since we compare with them, we should keep the same level of theory. However, although this could affect the energies of the CT minima the global shape of the TAS, on which we focus here, is maintained (see Figure in the attached file). Therefore, our analysis is not affected by this fact.
Transient absorption spectrum computed at equilibrium and non-equilibrium for the min-CT of anti-syn AG.

Reviewer 3 Report
Comments and Suggestions for Authors
See attached

Author Response
Line 126-131: In this connection, the author should cite the paper “dx.doi.org/10.1021/jp202280g | J. Phys. Chem. B 2011, 115, 8009–8013”. In this work, the authors identified the impurity and estimated the level of contamination in synthetic oligodeoxynucleotides as well as they showed how the contamination affected the results.
We cited this paper in the material section “Materials and methods” where we wrote:
"AG and GA dinucleotides, purified by desalting and, subsequently, by reverse phase HPLC, were purchased by Eurogentec. The corresponding MALDI-TOF spectra, shown in Figure S1 in Supplementary Materials (SM), indicate that both impurities and monomeric constituents have been efficiently removed. We note that electron paramagnetic resonance measurements detected other contaminants in commercially available oligonucleotides [1]."
The major issue of this work is computational techniques (see lines 175-176). TD-DFT underestimates the spin-orbit coupling term (see, Chem. Eur. J. 10.1002/chem.202100787). This leads to shifting the energy by -0.65 eV.
As explained in reference [2], the of the computed absorption spectra in respect to the experimental ones is not due to the absence of the spin-orbit coupling (SOC) term in the calculations, but, in part, it arises from the absence of vibronic effects. We added the following sentence in the “Computational details”:
"The computed spectra are compared with the experimental ones after shifting their energy by -0.65 eV, part of this difference arising from the absence of vibronic effects in our computations [2]."
Furthermore, although SOC may become important in epigenetic bases as described in the reference mentioned by the reviewer (Chem. Eur. J. 10.1002/chem.202100787) or monomers in gas phase [3], their role in systems involving canonic nucleobases in solution is less important.
To account for this, one has to take individual water (7-10) molecules around the dideoxynucleoside and repeat all the calculations. The role of the surrounding solvent (here water) in the DNA charge transfer processes has been illustrated (CHAPTER 31 Gamma- and Ion-beam DNA Radiation Damage: Theory and Experiment. DNA Damage, DNA Repair and Disease: Volume 2, Edited by Miral Dizdaroglu and R. Stephen Lloyd, The Royal Society of Chemistry 2021, Published by the Royal Society of Chemistry, www.rsc.org). Therefore, I recommend major revision and would like to review the manuscript again.
We thank the reviewer for having indicated this Chapter, which allowed us to place the ionization process in a more general frame. We cited it at the very beginning of the Introduction.
"Guanine radical cations, which are precursors to oxidative DNA damage, may be generated through an ionization process triggered by ion beams or electromagnetic radiation [4]."
However, the main topic of the above Chapter is the radical reactivity, while our present study focuses on the excited state relaxation. And we judge that, in this case, the use of the simpler continuum model is preferable. As a matter of fact, the definition of the first solvation shell for a dinucleotide is not straightforward and can introduce artifacts. Moreover, as shown in the Figure below, the solvation model hardly affects the shape of the computed TAS, which are used to interpret the experimental results.
In the discussion we make the precision:
..although the comparison between computed and experimental spectra in the present study is not intended to be quantitative, the spectra of different systems computed following the same methodology reveal interesting trends.
Transient absorption spectrum computed at equilibrium and non-equilibrium for the min-CT of anti-syn AG (see attached file)

Reviewer 4 Report
Comments and Suggestions for Authors
The manuscript by Petropoulos et al. describes the theoretical and experimental findings related to the ultrafast dynamics involving two specific dinucleotides, 5’-AG-3’ and 5’-GA-3’. The paper is well-written and the science is sound. I recommend the publication of this work.
I only have a few minor comments:
1) Figure 4, the structures related to the GA conformers should be reversed, otherwise they are exactly like those of AG.
2) Page 10, line 354: missing units for the lifetime 1.4±0.2.
3) Page 12, lines 395 and 398: if I am not mistaken, it should be Figure 9b, instead of 8b.
4) There is a typo in the SM, Figure S2, caption: 110 ps vs 112 ps (manuscript).

Author Response
I only have a few minor comments:
- Figure 4, the structures related to the GA conformers should be reversed, otherwise they are exactly like those of AG.
We modified Figure 4, which is now Figure 5.
- Page 10, line 354: missing units for the lifetime 1.4±0.2.
We added the units (ps).
- Page 12, lines 395 and 398: if I am not mistaken, it should be Figure 9b, instead of 8b.
This was indeed an error. But the figure numbering changed because we separated the sections “Results” and “Discussion”.
- There is a typo in the SM, Figure S2, caption: 110 ps vs112 ps (manuscript).
We corrected this typo.

Round 2
Reviewer 3 Report
Comments and Suggestions for Authors
This manuscript is sufficiently improved and the authors have addressed my concerns/points. However, in the future efforts, I request the theory expert (Roberto Improta) to address the DFT method development so that the shift of -0.55 eV could be avoided in future.
I recommend "accept in present form".